# Electroencephalogram synchronization measure as a predictive biomarker of Vagus nerve stimulation response in refractory epilepsy: A retrospective study

Venethia Danthine[1]*, Lise Cottin[2], Alexandre Berger[1,5], Enrique Ignacio Germany Morrison[1,4], Giulia Liberati[1,6], Susana Ferrao Santos[1,3], Jean Delbeke[1], Antoine Nonclercq[2], Riëm El Tahry[1,3,4]

1 Institute of NeuroScience (IoNS), Université Catholique de Louvain, Ottignies-Louvain-la-Neuve, Belgium,
2 Bio- Electro- And Mechanical Systems (BEAMS), Université Libre de Bruxelles, Brussels, Belgium,
3 Department of Neurology, Cliniques Universitaires Saint Luc, Woluwe-Saint-Lambert, Belgium, 4 Walloon Excellence in Life Sciences and Biotechnology (WELBIO) department, WEL Research Institute, Wavre, Belgium, 5 Sleep and Chronobiology Lab, GIGA-Cyclotron Research Center-in Vivo Imaging, University of Liège, Liège, Belgium, 6 Institute of Psychology (IPSY), Université Catholique de Louvain, Ottignies-Louvain-la-Neuve, Belgium

* Venethia.danthine@uclouvain.be

**Data Availability Statement:** Data cannot be shared publicly because of ethical restrictions. Data are available from the Ethics Committee of Catholic

## Abstract

There are currently no established biomarkers for predicting the therapeutic effectiveness of Vagus Nerve Stimulation (VNS). Given that neural desynchronization is a pivotal mechanism underlying VNS action, EEG synchronization measures could potentially serve as predictive biomarkers of VNS response. Notably, an increased brain synchronization in delta band has been observed during sleep–potentially due to an activation of thalamocortical circuitry, and interictal epileptiform discharges are more frequently observed during sleep. Therefore, investigation of EEG synchronization metrics during sleep could provide a valuable insight into the excitatory-inhibitory balance in a pro-epileptogenic state, that could be pathological in patients exhibiting a poor response to VNS. A 19-channel-standard EEG system was used to collect data from 38 individuals with Drug-Resistant Epilepsy (DRE) who were candidates for VNS implantation. An EEG synchronization metric–the Weighted Phase Lag Index (wPLI)—was extracted before VNS implantation and compared between sleep and wakefulness, and between responders (R) and non-responders (NR). In the delta band, a higher wPLI was found during wakefulness compared to sleep in NR only. However, in this band, no synchronization difference in any state was found between R and NR. During sleep and within the alpha band, a negative correlation was found between wPLI and the percentage of seizure reduction after VNS implantation. Overall, our results suggest that patients exhibiting a poor VNS efficacy may present a more pathological thalamocortical circuitry before VNS implantation. EEG synchronization measures could provide interesting insights into the prerequisites for responding to VNS, in order to avoid unnecessary implantations in patients showing a poor therapeutic efficacy.

University of Louvain – Saint-Luc University Hospital (contact via commission.ethique-saintluc@uclouvain.be) for researchers who meet the criteria for access to confidential data.

**Funding:** VD is supported by a Fond de la Recherche Scientifique (F.R.S.-FNRS) with a FRIA grant. EIGM is funded by the Walloon Excellence in Life Sciences and Biotechnology (WELBIO) department of the WEL Research Institute (X.2001.22). RET is funded by the Walloon Excellence in Life Sciences and Biotechnology (WELBIO) department of the WEL Research Institute (X.2001.22) and the Queen Elisabeth Medical Foundation (QEMF). The funders had no role in study design, data collection and analysis, decision to publish, or preparation of the manuscript.

**Competing interests:** I have read the journal's policy and one of the authors of this manuscript have the following competing interests: Antoine Nonclercq reports being a shareholder of Synergia Medical SA, Belgium. The other authors have declared that no competing interests exist. This does not alter our adherence to PLOS ONE policies on sharing data and materials.

## 1. Introduction

Epilepsy is a neurological disease characterized by an unusual excitation of neurons, leading to a hypersynchronous state and cerebral dysfunction, resulting in the occurrence of seizures. If the epilepsy is refractory, patients are referred to an epilepsy center for a presurgical evaluation. If resective surgery is not possible, Vagus Nerve Stimulation (VNS) can be offered as an add-on treatment. While response to VNS may increase over time [1–3], after 2 to 4 years of VNS therapy, it is known that about 60% of implanted patients will experience a 50% seizure reduction—a group referred to as Responders (R) [1–3]. However, the mechanisms of action remain incompletely known, and until now, no predictive biomarkers can establish individual VNS response before the implantation. Using various recording techniques—including low-density scalp EEG [4–6] or stereo-EEG [7], it has been suggested that VNS reduces epileptic susceptibility by decreasing excitability of the cortex. Indeed, VNS is thought to partially exert its anti-seizure effects through the activation of widespread thalamocortical connections via the activation of the Locus Coeruleus (LC)—noradrenergic system [8], receiving projections from the Nucleus Tractus Solitarius (NTS) [9].

In the last decade, there has been a growing interest in exploring the functional interconnections among brain regions using functional connectivity (FC) analyses [10]. FC analyses highlighted pathological functional organizations in an array of neurological diseases [11–14], including epilepsy [15, 16].

In epilepsy research, several studies used EEG-derived FC metrics with the aim of 1) improving diagnosis [17, 18], 2) predicting the occurrence of seizures [19], and/or 3) assessing the effect of different therapies [5, 20, 21]. While various FC measures exist, the metrics focusing on the coupling of signal phases are known as phase synchronization measures [22]. Previous studies that used EEG-based phase synchronization measures (weighted Phase Lag Index–wPLI, Phase Lag Index–PLI, or Phase Locking Value—PLV) revealed a desynchronization effect with acute VNS administration [4–6].

In particular, one study has specifically noted that this desynchronization occurs during sleep [6]. This interest in sleep stems from sleep physiology, particularly stage 2 of sleep. Indeed, during NREM sleep, thalamo-cortical oscillatory phenomena–including sleep spindles and, to a lesser extent, K-complexes—are observed, and are typically found during sleep stage N2 [23–25]. These oscillations have been theorized to mirror certain thalamocortical circuits implicated in the generation of pathological epilepsy oscillatory phenomena, such as spike-wave discharges [26, 27].

However, studies investigating EEG-derived phase synchronization measures to predict VNS response before the implantation remain limited. A study conducted in 88 pediatric patients with drug-resistant epilepsy (DRE) during wakefulness built a classification model based on clinical data as well as synchronization features (wPLI, PLI and PLV). The model reached a classification accuracy of 75.5% with a precision of 80.8% in a discovery cohort (N = 70). In the validation cohort, the prediction model demonstrated an accuracy of 61.1% (N = 18). When only the EEG metrics were considered independently from clinical data, a higher PLI was observed in R compared to non-responders (NR, < 50% reduction in seizure frequency) for the high-beta band, while wPLI values did not differ between both groups [28]. A second study investigating awake EEG phase synchronization via the PLI before and after implantation could not discriminate R from NR based on frequencies between 0.1 Hz and 30 Hz [29]. Another study that investigated the power spectrum of EEG bands, found an abnormal reactivity in the alpha rhythm under photic stimulation and hyperventilation before implantation in R. The authors suggested that differential neuronal excitability and synaptic transmission could exist among patients, and could influence the therapeutic response [30].

As VNS acts through thalamocortical activation [31–35] sharing similar anatomical sites with sleep processes [23–25], investigation of FC metrics during sleep could give an insight into the prerequisites for responding to VNS. A previous work from our laboratory found a higher desynchronization in the theta band during sleep in R compared to NR [6]. As this difference was state-specific, these results could indicate that VNS may exert its anti-seizure effects differently between sleep and wakefulness.

Based on these findings, one could hypothesize that the sleep EEG data obtained prior to the implantation might offer predictive features into VNS response. Moreover, it could be hypothesized that NR may exhibit an inherently disordered sleep pattern reflected in altered EEG phase synchronization, partly due to pathological thalamocortical circuitry. Indeed, polysomnography (PSG) findings suggest that sleep quality correlates with a less severe epilepsy and better clinical effects in DRE [36, 37].

Consequently, in order to develop predictive biomarkers of VNS response, this study aims to extract EEG-based synchronization metrics before VNS implantation and compare them during sleep and wakefulness and between R and NR. This study constitutes the first investigation using the wPLI–a metric known to be more sensitive to true neural synchronization due to its insensitivity to noise [38, 39]—to investigate EEG phase synchronization in DRE patients during wakefulness and sleep before the implantation of a VNS device.

## 2. Materials and methods

### 2.1 Study population

We reviewed the medical records of the patients retrospectively starting November 1st 2021. Medical records of patients from the epilepsy surgery database of Saint Luc University Hospital were screened between January 1st, 2014 and January 1st, 2019. Inclusion criteria for the study were: (i) diagnosed with drug-resistant epilepsy (DRE); (ii) received a cervical VNS implant (DemiPulse Model 103 or DemiPulse Duo Model 104, AspireHC Model 105 or AspireSR Model 106; LivaNova, Inc., London, United Kingdom LivaNova, London, UK) between 2015 and 2021; (iii) aged between 18 and 75 years; (iv) underwent a video-EEG monitoring of at least 48h with a simultaneous electrocardiogram (ECG) before VNS implantation which was accessible for analysis. Patients with seizure reduction after VNS implantation concomitant to a change of medication have been excluded. Of the 110 patients implanted for the first time, 38 met all the inclusion criteria. Patients were classified either as R ($\geq$ 50% seizure frequency reduction) or as NR ($<$ 50% seizure frequency reduction) based on the assessment of clinical records one year after the implantation. Partial responders (PR– 30–50% seizure frequency reduction) were included in the NR group for the analyses. Of the 38 patients included, 27 kept seizure diaries from which it was possible to calculate the exact seizure frequency reduction instead of performing a binary R/NR classification.

Video-EEG recordings were performed at Saint-Luc University Hospital or William Lennox Center (Ottignies, Belgium), with a Deltamed® system (Natus Europe, Paris, France). Signals were digitized at a sampling rate of 256 Hz. Nineteen scalp electrodes ("Fp1", "F3", "F7", "C3", "T3", "T5", "P3", "O1", "Fp2", "F4", "F8", "C4", "T4", "T6", "P4", "O2", "Fz", "Cz", "Pz") were used and positioned according to the 10–20 system. EEG traces were reviewed for each patient to ensure that no seizure or status epilepticus occurred at least 3h preceding the epochs selection, since it could significantly alter brain synchronicity [40, 41].

A minimum of 7 to a maximum of 10 EEG epochs of 10 seconds were collected in two different states: (i) calm wakefulness, without any evident motor activity (including noticeable eye blinks and movement artifacts as well as major interictal epileptic activity); (ii) stage 2

NREM (N2) sleep. Sleep epoch selection was performed upon visual analysis, using the presence of sleep spindles and/or K-complexes as determinants for the N2 stage labeling.

EEG processing was carried out in MATLAB R2021a (Mathworks, Natick, USA), using in-house developed scripts and the Letswave 6 toolbox (UCLouvain, Brussels, Belgium) [6].

EEG pre-processing was completed by re-referencing to the common average [42] of the 19 selected EEG channels. Band-pass filtering (4th-order Butterworth filter) was applied to keep the frequencies of interest between 0.5 and 30 Hz.

The study was conducted after approval by the local Ethics Committee (Commission d'Ethique Hospital-Facultaire of Saint-Luc University Hospital/UCLouvain) (2018/07NOV/ 416). The Ethical Committee authorized the realization of the present retrospective study, as patients included agreed upon sharing data for academic studies at their first contact with Saint-Luc University Hospital. Data were collected, anonymized, and processed according to General Data Protection Regulation (GDPR) guidelines.

## 2.2 Connectivity analysis: EEG synchronization measures

As we introduced earlier, amongst the various existing connectivity measurements, we have chosen to focus on the phase-synchronization measures. These measures are used to quantify phase synchronization between EEG signals [43, 44]. The PLI is one of the most well-known metrics. This metric estimates the asymmetry of distribution of phase differences between two signals and is partially corrected for volume conduction [45]. The wPLI improves the PLI by weighting the phase differences by their magnitude, reducing the risk of bias introduced by small noise perturbations and making the wPLI more sensitive to detect true neural synchronization [38, 39].

This measure is based on the cross-spectrum between EEG signals, representative of a frequency domain cross-correlation, and computed as follows:

$$X = x(t)y(t)e^{\Delta\varphi(t)}$$

Where $\Delta\varphi(t)$ is the phase difference between the signals at time $t$.

The wPLI is obtained by computing the cross-spectrum on 1-s time windows, extracting a quantity of interest from it, and then averaging on these small-time windows via the expectation operator. This is described in the following equation:

$$wPLI = \frac{|E[|Im(X)|.\text{sgn}(Im(X))]|}{E[|Im(X)|]}$$

Where:

- $E[]$ is the expectation operator over time

- $Sgn$ is the signum function

- $Im()$ is the imaginary component

As it is based on the cross-spectrum, the wPLI is defined in the frequency domain. Its values for the classical narrow frequency bands of interest, including delta (0.5–4 Hz), theta (4–8 Hz), alpha (8–13 Hz), beta (13–30 Hz), and broadband (0.5–30 Hz) are obtained by averaging on the subset of frequencies constituting these ranges.

**2.2.1 Whole brain analysis.** First, the connectivity analysis is considered at the whole-brain level by investigating all possible pairwise combinations of EEG signals amongst the 19 electrodes. For each patient, this results in a 19 x 19 matrix of wPLI values for each state (awake or sleep) for every frequency band of interest.

**Fig 1. Regional electrode pooling for connectivity analysis.** Seven different regions were defined: three per hemisphere (frontal, parietal and occipital) and one central region [6].

The Dijkstra algorithm was used to highlight the connections on the shortest paths between any pair of electrodes (described here as nodes), considered as the most relevant. These connections were kept in the final network, while the others were put to zero [46]. Finally, these matrices were averaged across all channel pairs to obtain one mean global wPLI per state and frequency band for each patient.

**2.2.2 Topographic analysis.** We performed regional analyses by pooling specific subgroups of electrodes and computing connectivity values restricted to those regions. Seven subgroups were established (Fig 1). Each subgroup's mean wPLI was computed and compared per group, per state, and frequency band of interest following the previous analysis.

## 2.3 Statistical analysis

Statistical analyses were performed using R® (version 4.1Ff.2). Demographic and clinical characteristics of the study population were statistically compared between the R and NR groups, using the Mann–Whitney U-test for the continuous variables and Fisher's exact test for the categorical variables.

To avoid multicollinearity problems inherent in variables, the Variance Inflation Factor (VIF) was computed for the predictors included in each model, and VIF values were lower than 5 (i.e., a low correlation between the predictors). Additionally, the Cook distance was computed to detect potential outliers. This technique did not detect any outliers.

A Linear Mixed Model (LMM) was used to analyze wPLI (used as the dependent variable in the LMM), in terms of state (awake/sleep), VNS response (R/NR), and interaction between the two (state/future response), using the 'lmer' function (R package 'lme4'). Covariates were added to the model to control for age, sex, number of anti-seizure medications (ASM), type of epilepsy (focal or generalized), and benzodiazepine intake. The subject ID was used as a random variable to tease out the influence of the inter-subject variability. To confirm the interest of a model using a random effect, it is necessary to check whether the likelihood of the model is improved when the random effect is added compared with a model without a random effect, using the REstricted Maximum Likelihood (REML). The normality of residues of the model was verified with the Shapiro test. A post-hoc analysis using Chi-square tests ('testInteraction' function in R, package 'phia') was performed to limit false positive findings, using False Discovery Rate FDR correction for multiple comparisons [47]. The level of significance was set at $p < 0.05$.

In order to investigate the continuous measure of seizure frequency reduction from baseline in both states, a multiple linear regression (using the 'lm' function in R, package 'Stat') was built in the subgroup of patients for which the exact seizure frequency reductions were available using the backward method. The relationship between wPLI and future percentage of seizure reduction (after one year of VNS) was investigated, correcting for the number of anti-

seizure medications (ASM) and the patient's age at time of the EEG acquisition. The level of significance was set at p<0.05. Visual inspection, R2, and Root-Mean-Squared Error (RMSE) of the models were assessed as a measure of the suitability of the fit.

# 3. Results

## 3.1 Study population: Clinical characteristics

Thirty-eight patients (20 females and 18 males; 12 R and 26 NR—including 13 PR) were eligible for inclusion after a review of our clinical database. A comparative table of demographics and clinical features of the R and NR groups can be found in Table 1. No statistical significance was found between groups, except for the type of epilepsy.

## 3.2 Connectivity analysis: EEG synchronization measures

**3.2.1 Whole brain analysis.** The usefulness of a random variable in the model has been confirmed in two bands: delta **(p = 0.006\*\*)** and alpha **(p = 0.04\*)** bands.

In the delta band analysis, the LMM revealed that the state variable exhibited a significant effect on wPLI **(p = 0.016\*)**, with lower wPLI values found in sleep compared to wakefulness (S1 Table). On the other hand, the response variable showed no significant impact on wPLI (p = 0.29), indicating no difference in wPLI between responders (R) and non-responders (NR). Moreover, the interaction between State and Response was not significant (p = 0.27), suggesting that our model cannot distinguish between R and NR within each state in delta band. Utilizing LMM allows us to conduct post-hoc tests, known as contrasts, within the model. These contrasts enable specific comparisons of interest between levels of categorical variables in the LMM. In the delta band model, contrasts revealed that higher wPLI values

**Table 1. Demographic and clinical characteristics of the study population.**

| Variables | R (n = 12) Mean (SD) | NR (n = 26) Mean (SD) | p-value |
|---|---|---|---|
| Sex* | 7F – 5M | 13F – 13M | 0.92 |
| Age (years) | 37.9 (12.8) | 37.4 (13.6) | 0.92 |
| Age of Epilepsy onset (years) | 16 (11.07) | 14.24 (16.5) | 0.71 |
| Epilepsy duration (years) | 21.18 (16.2) | 23.52 (16.5) | 0.70 |
| **Epilepsy type** | | | <2.2×10−16 |
| Foca | 12 | 18 | |
| Generalized | 0 | 7 | |
| Both | 0 | 1 | |
| **Localization** | | | 0.66 |
| Temporal (T) | 7 | 7 | |
| Extra-temporal (ET) | 4 | 16 | |
| Both | 1 | 3 | |
| **Mean no. of ASMs used** | 3 (1.17) | 2.6 (0.77) | 0.24 |
| No. using benzodiazepines | 3/12 | 3/26 | 0.38 |
| No. using antidepressants (SSRI, TCA) | 3/12 | 4/26 | 0.38 |
| **Epilepsy lateralization** | | | 0.75 |
| Right | 1 | 4 | |
| Left | 6 | 12 | |
| Bilateral | 5 | 10 | |

F: Females; M: Males; ASM: Anti-Seizure Medication; SSRI: Selective Serotonin Reuptake Inhibitor; TCA: Tricyclic Antidepressant Agent

*Sex assigned at birth

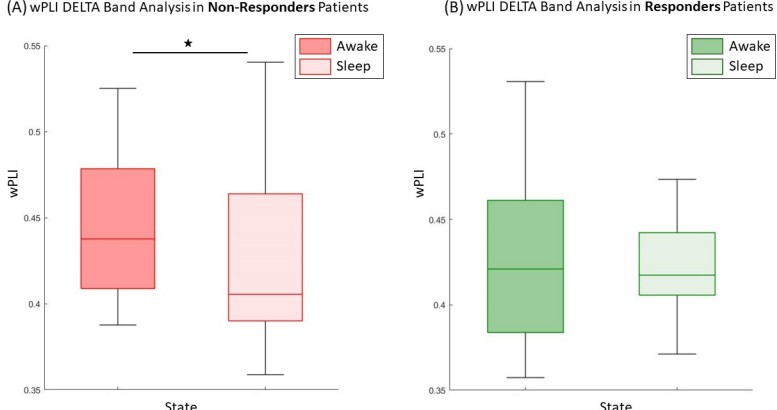

**Fig 2. Boxplots of the wPLI in the delta band. (A)** NR patients demonstrated a higher wPLI in wakefulness compared to sleep while **(B)** no difference is found between states in R patients. ⋆ Significant result at the level p<0.05 in post-hoc comparison".

were observed during wakefulness compared to sleep, but only for non-responders **(p = 0.022\*)**. No significant difference in wPLI between wakefulness and sleep for responders was found (p = 0.71). Please refer to Fig 2 for a visual representation.

In the analysis of the alpha band, a significant effect of the state on wPLI **(p = 0.044\*)** is found, while neither the response (p = 0.57) nor the interaction showed significant impacts (p = 0.17). We did not observe any significant results in the contrast analysis. However, a trend emerged during sleep between responders (R) and non-responders (NR) (p = 0.058), with NR exhibiting a higher wPLI compared to R, while no trend is found in wakefulness (p = 0.57). Please refer to Fig 3 for a visual representation.

More detailed information about the LMM can be found in S1 and S2 Tables.

No significant difference was found between the R and NR groups using an LMM in the other bandwidths.

To mitigate biases inherent of a binary classification and investigate in more detail the trend found in alpha band in sleep, our aim was to address the specific percentage of seizure reduction through a regression analysis.

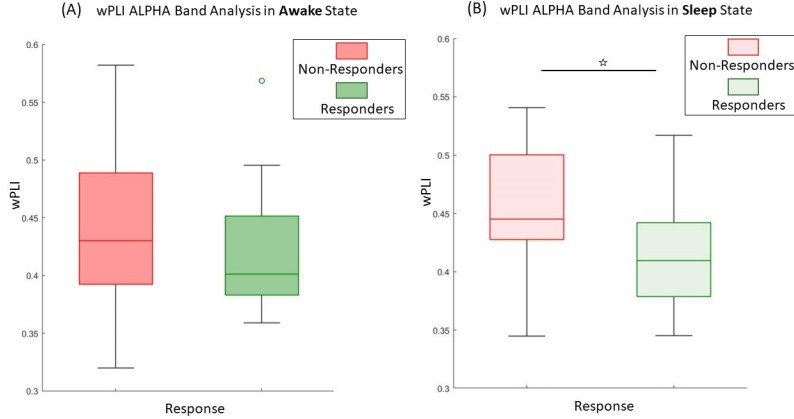

**Fig 3. Boxplots of the wPLI in the alpha band. (A)** In wakefulness, no difference was found between groups while **(B)** in sleep, a higher wPLI is found in NR group compared to R. ☆ Trend of significance in post-hoc comparison (p = 0.058).

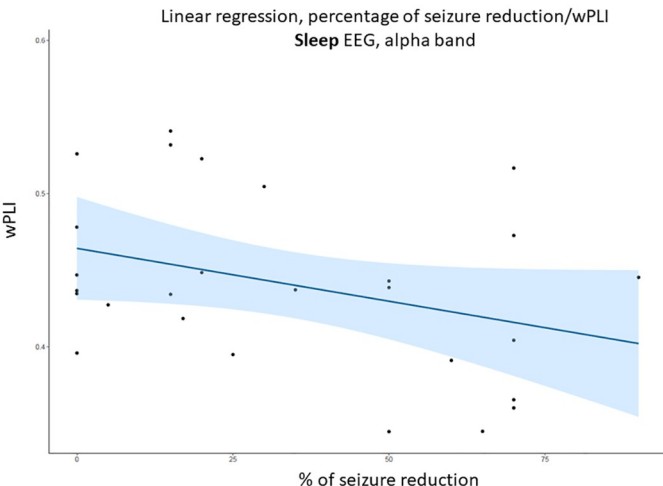

**Fig 4. Linear regression between wPLI in the alpha band during sleep and the percentage of seizure reduction.**
During sleep, a lower wPLI was observed in patients with a higher reduction in seizure frequency.

For patients in whom the exact percentage of seizure reduction was available (N = 27), the multiple linear regression in the alpha band during sleep revealed a significant association between percentage of seizure reduction (**p = 0.014***) and global wPLI, indicating a decrease in wPLI among patients exhibiting a stronger response to VNS (i.e., greater seizure reduction). Secondly, age demonstrated a significant correlation with wPLI (**p = 0.005****), revealing higher wPLI levels among older patients. Interestingly, the number of antiseizure medications (ASMs) did not yield a significant effect on wPLI but contributed to enhancing the accuracy of the model (adjusted R-squared: 0.34; statistical comparison of the linear regressions can be found in S3 Table). For visual representation, a linear regression between the wPLI in the alpha band during sleep and the percentage of seizure reduction is shown in Fig 4.

**3.2.2 Topographic analysis.** Considering that the results were significant only in the alpha and delta bands, we focused on these two bands only for the regional sub-analysis (Fig 1). No result remained significant after FDR correction, neither between groups (R-NR) nor between states (awake-sleep).

## 4. Discussion

Using wPLI as a connectivity metrics [22], this study attempts to identify predictive markers of VNS response by analyzing pre-implantation EEG data during wakefulness and sleep. Despite the absence of a clear predictive marker for VNS efficacy, this study points to certain physiological factors that appear to impact the therapeutic response. Indeed, we found a higher wPLI (i.e. may be interpreted as higher FC) in the delta band during wakefulness compared to sleep, specifically in NR. Conversely, a higher reduction in seizure frequency with VNS is correlated with reduced alpha band wPLI during sleep before VNS implantation. Although the physiological interpretation of these findings remains challenging, our results may reflect more severe pathological brain alterations in NR before the implantation.

Only a few studies analyzed phase synchronization measurements to predict VNS response. A first study using PLI found no difference between R and NR in the range of 0.5–30 Hz [29]. Another study conducted in children before the implantation observed an increased global PLI in R group in wakefulness, specifically in the high-beta band, compared to the NR group.

While other synchrony measures, such as PLV and wPLI, showed no differences between the two groups during wakefulness, EEG-derived synchronization metrics were not investigated during sleep [28].

Interestingly, no differences between wakefulness and sleep was found among our patients in global wPLI for the alpha band activity. These findings were somewhat unexpected, considering the existing literature that underscores the significance of alpha rhythm in consciousness in healthy patients. At first glance, our results seem to be inconsistent with previous findings. In healthy participants a higher connectivity within the alpha band was observed during wakefulness compared to sleep. During sleep, the enhancement of wPLI was found in the delta band instead [48–50].

However, studies focusing on epileptic patients beyond the scope of VNS response have suggested that an increased connectivity may reflect pathological brain states [51–53]. Using high-density EEG and resting-state functional MRI, Carboni *et al.* compared global efficiency derived from partial directed coherence EEG measures between epileptic patients and healthy subjects. They identified an increased connectivity in epileptic patients compared to healthy subjects. The authors suggested that an increased global efficiency could reflect a more extensive pathological (epileptic) network within the brain [51].

Similarly, Nayak *et al.* investigated an EEG phase synchronization index (SI) within awake and sleep EEG recordings. Their findings revealed a higher SI within the delta and theta frequency bands amongst epileptic patients compared to controls across both wakefulness and sleep states, irrespective of the epilepsy subtype. These results suggested that an augmented synchronization could serve as a distinctive trait of epileptogenic brain networks [53].

Likewise, using mutual information to compute different graph measures (including global efficiency, characteristic path length, average clustering coefficient, and modularity), Davis *et al.* found a higher connectivity in almost all bands during sleep (including the alpha band) in children with tuberous sclerosis complex who will develop epileptic spasms. Hence, the authors suggested that an over-connectivity may reflect a more pathological network [52]. In addition, the study of Brazdil *et al.* pointed out the potential importance of alpha rhythm as a VNS response predictor in the awake condition after photic stimulation and hyperventilation [30]. Considering the association between thalamocortical circuitry, their significance during sleep (especially in stage 2) [24, 54], and the alpha rhythm [55], it is plausible that this frequency band could potentially serve as an indicator of abnormalities within these circuits, even during sleep stage 2.

In line with these studies, our results indicate an inverse relationship between the wPLI and the exact number of seizure reduction after VNS implantation. Overall, our results support the hypothesis that patients with a more severely affected epileptic network are less likely to respond to VNS.

Other studies have also investigated the prediction of VNS response using other connectivity measures or techniques. Kim *et al.* employed the preoperative EEG-based Direct Transfer Function (DTF) to characterize brain connectivity in DRE patients compared to healthy subjects. While a similar connectivity profile was found between R to VNS and healthy subjects, NR showed a higher difference with the healthy population [56]. In addition to EEG, pre-implant MEG data has also been used to define response before VNS implantation. A study used PLV to compute graph measures (e.g., modularity, transitivity, and characteristic path length) [57]. Another study used machine learning techniques to identify good candidates for VNS implantation using a combination of structural (using diffusion tensor imaging) and functional (using resting-state MEG) metrics [34]. These techniques showed a greater structural connectivity in left thalamocortical, limbic, and association fibers and a greater functional connectivity in a network composed of the left thalamic, insular, and temporal regions [34].

Both studies underline a closer connectivity pattern between future R and controls compared to NR [34, 57]. Finally, using resting-state functional MRI data and a machine learning approach, Ibrahim *et al.* showed an increased connectivity of the thalami to the anterior cingulate cortex and left insula that was associated with greater VNS efficacy [33]. These latest studies demonstrate the value of studying deep brain regions for predicting VNS response, which is not possible with scalp EEG. This could partly explain the lack of highly significant results in our study.

Finally, in post-implantation studies, phase synchronization measures have shown their value as a marker of response to VNS. In this way, utilizing the PLI as a connectivity measure, Bodin *et al.* discovered that during wakefulness, R exhibited reduced synchrony in delta and alpha bands compared to NR [4]. Notably, the ON periods were consistently associated with lower PLI values than the OFF periods in R and NR groups. These results reflect the acute desynchronizing effect of VNS. This is consistent with the results of Sangare *et al.*, who reported a lower PLI in delta, theta, and beta bands acutely during the ON period compared to the OFF period in R only [5]. In addition, Vespa *et al.* computed an OFF/ON ratio based on the averaged whole-brain wPLI in wakefulness vs. stage 2 sleep and found a greater OFF/ON wPLI ratio within the theta band in R compared to NR [6].

## 4.1 Limitations and future perspectives

First, it is important to mention that our selection of N2 sleep epochs was based solely on the EEG data available, without including PSG, as per AASM (American Academy of Sleep Medicine) guidelines. While we cannot exclude the possibility that transitional N1 or N3 periods may have been partially included in the chosen EEG traces, exploring other sleep states (e.g., NREM N1-N3 and REM sleep) would be interesting. Indeed, some studies have indicated that VNS can also influence REM sleep [58, 59], and there may be links between REM sleep mechanisms and certain forms of epilepsy [26, 60, 61]. It must be noted that we were unable to rule out the contamination of selected epochs entirely by epileptiform activity, as concomitant intracranial EEG recordings were not available. Such epileptiform activity might have influenced background EEG rhythms and connectivity.

Additionally, in our dataset, a significant distinction arises between responders (R) and non-responders (NR) concerning epilepsy type. Nevertheless, we maintain that this factor does not impact our primary findings. Indeed, this imbalance of epilepsy type between the groups could be due to our relatively low sample size, while previous studies did not find association between VNS response and epilepsy type [4, 6]. Furthermore, Sangare *et al.* investigated the PLI difference between focal and generalized epilepsy and no difference was found [5]. Moreover, we have addressed this variable in our models to ensure its control in our statistical analyses. No effect of epilepsy type was found in the mixed model analysis or in the regression model analysis (S3 Table)."

We acknowledge that our study was performed retrospectively on a heterogeneous population in terms of type of epilepsy, age, gender, or medication. Furthermore, conducting subgroup analyses would have been impractical due to our study's relatively small sample size. Given the inherent patient-to-patient variability and the interest of following patient before and after the VNS surgery, we propose a longitudinal follow-up study to address the temporal variability of the wPLI and the impact of VNS on this measure. Finally, the evaluation of connectivity as limited by the fact that we only used a single connectivity measure, from one family. Comparing several connectivity or network measures from different families (based on different mathematical principles) might better reflect physiological functioning, allowing us to predict a potential VNS response.

## 5. Conclusion

This study aimed to find a predictor for clinical response to VNS in epileptic patients using EEG phase synchronization metrics. The results of this investigation show that DRE patients, in particular NR, have a greater functional connectivity in the delta band during wakefulness compared to sleep. In addition, a higher seizure reduction is correlated with a lower alpha band connectivity in sleep. Overall, our results support the hypothesis of a more pathological brain in NR, which could explain the lower therapeutic efficacy of VNS. However, although wPLI may help to establish VNS response after the implantation, this connectivity metric alone may not be adapted for predicting VNS response. Further studies focusing on a physiological explanation of the connectivity measures used are needed to explore the full potential of connectivity metrics.

## Supporting information

**S1 Table. Linear mixed models using wPLi as dependent variable in delta band with and without covariables.**
(DOCX)

**S2 Table. Linear mixed models using wPLi as dependent variable in alpha band with and without covariables.**
(DOCX)

**S3 Table. Models of multiple regression analysis using wPLI as a dependent variable in alpha band in sleep.**
(DOCX)

## Acknowledgments

This research has benefited from a statistical consultancy service with the Statistical Methodology and Computing Service, a technical platform at UCLouvain–SMCS/LIDAM, UCLouvain.

## Author Contributions

**Conceptualization:** Venethia Danthine, Alexandre Berger, Giulia Liberati, Jean Delbeke, Antoine Nonclercq, Riëm El Tahry.

**Data curation:** Venethia Danthine.

**Formal analysis:** Venethia Danthine, Lise Cottin, Alexandre Berger.

**Funding acquisition:** Venethia Danthine, Susana Ferrao Santos.

**Investigation:** Venethia Danthine, Giulia Liberati, Jean Delbeke, Antoine Nonclercq, Riëm El Tahry.

**Methodology:** Venethia Danthine, Giulia Liberati, Antoine Nonclercq, Riëm El Tahry.

**Project administration:** Venethia Danthine.

**Resources:** Venethia Danthine.

**Software:** Lise Cottin.

**Supervision:** Giulia Liberati, Jean Delbeke, Antoine Nonclercq, Riëm El Tahry.

**Validation:** Venethia Danthine, Lise Cottin.

**Visualization:** Venethia Danthine, Alexandre Berger.

**Writing – original draft:** Venethia Danthine, Lise Cottin, Riëm El Tahry.

**Writing – review & editing:** Venethia Danthine, Lise Cottin, Alexandre Berger, Enrique Ignacio Germany Morrison, Giulia Liberati, Susana Ferrao Santos, Jean Delbeke, Antoine Nonclercq, Riëm El Tahry.

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
