## [Decision Letter · Decision Letter 0]

23 Feb 2024

PONE-D-23-43902Electroencephalogram Synchronization Measure as a Predictive Biomarker of Vagus Nerve Stimulation Response in Refractory Epilepsy: A Retrospective StudyPLOS ONE

Dear Dr. Danthine,

Thank you for submitting your manuscript to PLOS ONE. After careful consideration, we feel that it has merit but does not fully meet PLOS ONE’s publication criteria as it currently stands. Therefore, we invite you to submit a revised version of the manuscript that addresses the points raised during the review process.

We look forward to receiving your revised manuscript.

Kind regards,

Ayataka Fujimoto

Academic Editor

PLOS ONE

“I have read the journal's policy and one of the authors of this manuscript have the following competing interests: Antoine Nonclercq reports being a shareholder of Synergia Medical SA, Belgium. The other authors have declared that no competing interests exist.”

4. In the online submission form you indicate that your data is not available for proprietary reasons and have provided a contact point for accessing this data. Please note that your current contact point is a co-author on this manuscript. According to our Data Policy, the contact point must not be an author on the manuscript and must be an institutional contact, ideally not an individual. Please revise your data statement to a non-author institutional point of contact, such as a data access or ethics committee, and send this to us via return email. Please also include contact information for the third party organization, and please include the full citation of where the data can be found.

Additional Editor Comments:

Reviewers perceive this manuscript positively, as it holds significant value. Please respond earnestly to the reviewers' comments, as the content is highly meaningful.

Reviewers' comments:

Reviewer's Responses to Questions

**Comments to the Author**

1. Is the manuscript technically sound, and do the data support the conclusions?

Reviewer #1: Partly

Reviewer #2: Yes

2. Has the statistical analysis been performed appropriately and rigorously? 

Reviewer #1: I Don't Know

Reviewer #2: Yes

3. Have the authors made all data underlying the findings in their manuscript fully available?

Reviewer #1: No

Reviewer #2: Yes

4. Is the manuscript presented in an intelligible fashion and written in standard English?

Reviewer #1: No

Reviewer #2: Yes

5. Review Comments to the Author

Reviewer #1: The study involved an analysis of 38 patients who underwent VNS implantation, aiming to identify predictive factors for VNS response by comparing EEG data between responders and non-responders. In particular, the authors assess brain synchronization by wPLI, considering that desynchronization is one of the mechanisms of VNS. Notably, the study uncovered an association between low wPLI during sleep and favorable seizure reduction. Given the absence of predictive biomarkers for VNS, this discovery holds substantial implications for clinical practice. However, the Introduction and Discussion are somewhat redundant, making it difficult to understand the main message of this study. Additionally, the presentation of results lacks adequate detail. Significant revisions are necessary before publication.

Comments (review invitation: Feb 12, 2024, and comment submission on Feb 16, 2024):

1. [Introduction/Discussion] Please describe the strengths or novelty of this research. What is the strong point compared to the previous studies with synchronization measurements such as PLI or wPLI?

2. [Introduction] The explanation of the mechanism of VNS seems redundant. There is no need to use several paragraphs to mention it.

3. [Introduction] It’s undesirable to use the term “hypothesize” to describe the ideas of other authors or articles. It would be better to use “hypothesize” only when describing your research hypothesis.

4. [Introduction] Please add more explanation of wPLI or PLI. Could you explain why these measurements are useful, citing past reports? It would clarify the importance of using these measurements in this study.

5. [Results] Due to the lack of explanation, it is hard to understand the results. An example is “For the delta band, the LMM revealed a significant effect of state (wakefulness/ sleep) (p=0.016*) but no effect of VNS response (p=0.29) nor an interaction between state and VNS response (p=0.27).”

・Does “a significant effect of state (wakefulness/sleep)” mean that there is a significant difference in wPLI between wake and sleep? Which states have higher wPLI?

・Does “no effect of VNS response (p=0.29)” mean that there is no difference in wPLI between VNS responders and non-responders?

・What does “an interaction between state and VNS response (p=0.27)” mean specifically?

Please describe the results in more detail or use Table, not just in this example but all results.

6. [Results] It would be better to add the information that a high wPLI means increased connectivity in addition to the results (Is my interpretation correct?). This is because many readers would be expected not to know what high wPLI means.

7. [Figure 2-4 legend] It would be nicer to describe an explanation of each result in Figure legend.

8. [Figure 3] It is not common to use symbols such as asterisks for results that are not significantly different. Please remove it as it is misleading.

9. [Discussion] Given that brain connectivity increases during sleep, wPLI during wake is considered to be lower than during sleep. Please discuss why the phenomenon opposite to the theory occurred.

10. [Discussion] The idea that more pathological connectivity is related to VNS non-responders seems to be inconsistent with the VNS mechanism. This is because VNS appears to alleviate the symptoms of epilepsy patients with high connectivity (=more pathological connectivity) through desynchronization. Please give a convincing explanation.

11. [General comment] Please double-check the grammar and logical development of the manuscript again.

12. [Minor comments]

・Brackets of “(Video-)EEG recordings were…” in 109 seems to be unnecessary.

・The use of the term “future” VNS response feels inappropriate in a retrospective study.

・Table 1: The use of “n°” is not common. It would be more understandable to use “no.” or “n”(italic).

・Table 1: It would be better to change “<2.2e-16” to “2.2×10-16” or “<0.001”.

・Results: What does the asterisk (for example, p=0.006*) mean? If unnecessary, please remove it.

Reviewer #2: Danthine et al. studied a predictor for clinical response to VNS in epileptic patients using EEG phase synchronization metrics. They mentioned DRE patients, in particular future non-responders, have a greater functional connectivity in the delta band during wakefulness compared to sleep.

Comments (invitation: February 11, 2024, and submission: February 18, 2024)

We want to congratulate the authors’ efforts. The manuscript is well written and easy to follow. None of the following comments are criticisms.

1) Abstract (line 28): Wouldn't the statement "especially in NR" be unnecessary since you later say that the difference between R and NR did not reach significance?

2) Abstract (line 31): I think that readers who solely read the abstract may not grasp the concept. Could you please elaborate on why future NRs may have a more pathological thalamocortical circuitry?

3) Introduction (line 86): It would be preferable to spell out the abbreviations for PSG when it is first mentioned.

4) Introduction (line 90-91): I understand what the authors are saying. And indeed, as the authors report, there was a negative correlation between the wPLI and the future percentage of seizure reduction after VNS implantation, only during sleep. In that case, it is natural to assume that changes have occurred in the postoperative EEG. Have you measured the postoperative EEG? If you did, please discuss this point.

5) Figure1 (line 168-169): The author may want to write “T4-C4” instead of “T4,Ca”.

6) Results (line 203 -204): R group is only focal epilepsy. I would appreciate it if you could discuss whether this affected the main finding (the difference in synchronization between two groups).

6. PLOS authors have the option to publish the peer review history of their article (what does this mean?). If published, this will include your full peer review and any attached files.

Reviewer #1: **Yes: **Keisuke Hatano

Reviewer #2: No

---

## [Author Response · Author response to Decision Letter 0]

25 Apr 2024

Please refer to the PDF document "Response to reviewers" that has been uploaded in the "Attach files" section.

---

## [Decision Letter · Decision Letter 1]

7 May 2024

Electroencephalogram Synchronization Measure as a Predictive Biomarker of Vagus Nerve Stimulation Response in Refractory Epilepsy: A Retrospective Study

PONE-D-23-43902R1

Dear Dr. Venethia Danthine,

We’re pleased to inform you that your manuscript has been judged scientifically suitable for publication and will be formally accepted for publication once it meets all outstanding technical requirements.

Kind regards,

Ayataka Fujimoto

Academic Editor

PLOS ONE

Additional Editor Comments (optional):

I have endorsed this version.

Reviewers' comments:

Reviewer's Responses to Questions

**Comments to the Author**

1. If the authors have adequately addressed your comments raised in a previous round of review and you feel that this manuscript is now acceptable for publication, you may indicate that here to bypass the “Comments to the Author” section, enter your conflict of interest statement in the “Confidential to Editor” section, and submit your "Accept" recommendation.

Reviewer #1: All comments have been addressed

Reviewer #2: All comments have been addressed

2. Is the manuscript technically sound, and do the data support the conclusions?

Reviewer #1: Yes

Reviewer #2: Yes

3. Has the statistical analysis been performed appropriately and rigorously? 

Reviewer #1: Yes

Reviewer #2: Yes

4. Have the authors made all data underlying the findings in their manuscript fully available?

Reviewer #1: No

Reviewer #2: Yes

5. Is the manuscript presented in an intelligible fashion and written in standard English?

Reviewer #1: Yes

Reviewer #2: Yes

6. Review Comments to the Author

Reviewer #1: I appreciate the effort you've put into revising the manuscript based on the feedback provided. Please modify the following as needed.

① In line 85 of the Introduction, the phrase “a higher wPLI desynchronization” appears confusing because a higher wPLI indicates more severe synchronization. It might be clearer to express it as “a higher desynchronization”.

② The parenthesis in “a (video-)EEG monitoring” is unnecessary.

③ There are some typos as follows:

・In line 32 of the Abstract, you mentioned “was found” twice in the same sentence (However, in this band, no synchronization difference was found in any state was found between R and NR).

・The parenthesis is not closed in line 272 in the Discussion (Using wPLI as a connectivity metrics (as wPLI is an accepted marker of connectivity (22),…). I think the sentence “as wPLI is an accepted marker of connectivity” is unnecessary because you have already mentioned the usefulness of wPLI in the Introduction. Please close the parenthesis or remove the sentence within the parenthesis.

・” seizure frequency” is better than “seizures frequency” in line 277 of Discussion (Conversely, a higher reduction in seizures frequency with VNS is correlated with reduced alpha band wPLI during sleep before VNS implantation).

Reviewer #2: The authors have replied sufficiently to all my comments. It is a very nice manuscript. Kazuki Sakakura

7. PLOS authors have the option to publish the peer review history of their article (what does this mean?). If published, this will include your full peer review and any attached files.

Reviewer #1: No

Reviewer #2: **Yes: **Kazuki Sakakura

---

## [Editor Report · Acceptance letter]

2 Jun 2024

PONE-D-23-43902R1 

PLOS ONE

Dear Dr. Danthine, 

I'm pleased to inform you that your manuscript has been deemed suitable for publication in PLOS ONE. Congratulations! Your manuscript is now being handed over to our production team.

Kind regards, 

on behalf of

Dr. Ayataka Fujimoto 

Academic Editor

PLOS ONE